# Rheumatic Diseases Development in Patients Treated by Anti-PD1 Immune Checkpoint Inhibitors: A Single-Centre Descriptive Study

**DOI:** 10.3390/life13040877

**Published:** 2023-03-25

**Authors:** Fulvia Ceccarelli, Francesco Natalucci, Licia Picciariello, Giulio Olivieri, Alessio Cirillo, Alain Gelibter, Vincenzo Picone, Andrea Botticelli, Fabrizio Conti

**Affiliations:** 1Arthritis Center, Reumatologia, Dipartimento di Scienze Cliniche Internistiche, Anestesiologiche e Cardiovascolari, Sapienza Università di Roma, 00185 Rome, Italy; 2Department of Radiological, Oncological and Pathological Science, Sapienza University of Rome, 00185 Rome, Italy; 3Division of Medical Oncology B, Policlinico Umberto I, Sapienza—Università di Roma, 00185 Rome, Italy

**Keywords:** immune checkpoint inhibitors, rheumatic diseases, systemic autoimmune diseases

## Abstract

The introduction of the so-called immune checkpoint inhibitors (ICIs) substantially changed the history of cancer therapy. On the other hand, they can induce the development of rheumatic immune-related adverse events (Rh-irAEs). In the scenario of a joint oncology/rheumatology outpatient clinic, we conducted a single-centre descriptive study to define from a laboratory, clinical and therapeutic point of view, rheumatic conditions developed during anti-PD1 treatment. The study included 32 patients (M/F 16/16, median age 69, IQR 16.5). According to the international classification criteria, eight patients could be classified as affected by Rheumatoid Arthritis, one by Psoriatic Arthritis, six by Polymyalgia Rheumatica, five by systemic connective tissue diseases (two systemic lupus erythematosus, two Sjögren’s syndrome, one undifferentiated connective tissue disease). The remaining patients were diagnosed as having undifferentiated arthritis or inflammatory arthralgia. The median interval between ICIs starting and the onset of symptoms was 14 weeks (IQR 19.75). Moving to treatment, the longitudinal observation revealed that all RA, PsA and CTD patients required the introduction of treatment with DMARDs. In conclusion, the growing use of ICIs in a real-life setting confirmed the possible development of different rheumatological conditions, further emphasising the need for shared oncology/rheumatology management.

## 1. Introduction

The modulation of the immune system has been extensively suggested as a valid therapeutic strategy to treat cancer patients. Thus, the so-called immune checkpoint inhibitors (ICIs) have been proven to be one of the most relevant advances in cancer therapy over the past decade [1]. These drugs had shown great efficacy with an increased response rate and survival in patients with different malignant diseases such as advanced-stage melanoma, renal cell carcinoma, head and neck squamous cell carcinoma, small-cell lung cancer and non-small-cell lung cancer and other solid tumors [2]. ICIs commonly used include anti-programmed cell death 1 (PD-1) and anti-programmed cell death 1 ligand 1 (PD-L1) antibodies. Under physiological conditions, PD-1 engagement by its ligand PD-L1 limits T-cell activation and maintains immune tolerance. In contrast, the expression of the inhibitory ligands in the surface of malignant cells leads to downregulation of the T-cell response, enabling tumour escape from immunosurveillance [3]. 

Therefore, ICIs have a beneficial role in activating tumour antigen-specific T cells, but they can also induce an aberrant activation of autoantigen-reactive T cells, leading to side effects that could resemble autoimmune diseases [4]. Consequently, a wide spectrum of immune-related adverse events (irAEs) has emerged, including rheumatic manifestations [5]. The incidence of rheumatic irAEs (Rh-irAEs) is worse characterised than other irAEs, due to lack of specific definitions of musculoskeletal manifestations in oncology clinical trials [4]. Given this aspect, overall, the prevalence of Rh-irAEs has been estimated from 0.4 to 16% [6,7]. So far, several studies focused on joint manifestations induced by ICIs; indeed, case series and retrospective reviews reported arthralgia in up to 43% of patients, whereas arthritis occurred in up to 7% [8]. Most cases are classified as undifferentiated arthritis (UA) and are seronegative for rheumatoid factor (RF) and anti-cyclic citrullinated peptide (ACPA). Nonetheless, in some cases a diagnosis of a defined rheumatic disease, such as rheumatoid arthritis (RA), psoriatic arthritis (PsA), or polymyalgia rheumatica (PMR) could be made. Although less frequently, the development of connective tissue diseases (CTDs), such as Systemic Lupus Erythematosus (SLE), has been described [5]. 

From an epidemiologic point of view, attention has been devoted to the interval between ICIs initiation and the onset of musculoskeletal symptoms. Data from the literature suggest that RA usually occurs after 1 month (from 3 days to 5 months), whereas undifferentiated oligoarthritis and polyarthritis develop around 3 months (1–9 months and 1 day–24 months, respectively) [9]. 

Regarding the treatment, the majority of patients treated with ICIs develop mild-to-moderate arthritis that generally responds well to non-steroidal anti-inflammatory drugs (NSAIDs) and low-dose glucocorticoids. About 30% of patients require disease-modifying antirheumatic drugs (DMARDs), and this kind of medicine is more frequent among patients who receive ICIs combination. In these cases, the most common choice is methotrexate [10]. A small number of patients with ICIs-induced arthritis might need treatment with agents targeting TNF or IL6, that are successfully used in some case series [11].

Taken together, these findings led to the increasing involvement of the rheumatologist in the management of ICIs treated patients. Thus, in the scenario of a joint oncology/rheumatology outpatient clinic, established from January 2017, we aimed at describing from a laboratory, clinical and therapeutic point of view rheumatic conditions developing during treatment with anti-PD1 immune checkpoint inhibitors. 

## 2. Materials and Methods

For the present study, we enrolled adult oncologic patients with a new onset of symptoms evocating rheumatic conditions which appeared after the introduction of anti-PD1 treatment. All patients were evaluated in the oncology/rheumatology outpatient clinic at the Sapienza University of Rome. Individuals with a previous diagnosis of rheumatic diseases were excluded. Patients’ clinical history was collected into a standardised computerised electronically filled form, including demographics, clinical and previous and current treatments information. We enrolled patients who were treated either by nivolumab at the dosage of 240 every 2 weeks or pembrolizumab at the dosage of 2 mg/kg every 3 weeks, according to the oncologic therapeutic schedule. 

From a rheumatologic point of view, patients were examined and evaluated for other symptoms suggestive of rheumatic diseases. 

In detail, guiding symptoms that were accurately researched from oncologists in order to refer the patients to our attention were inflammatory arthralgias, arthritis and/or other manifestations suspicious for rheumatic diseases (photosensitivity, malar rash, sicca syndrome, Raynaud phenomenon, psoriasis, aphthosis, serositis, haematological modifications, uveitis, purpura, thrombotic events).

The study was performed according to the protocol and good clinical practice principles of the Declaration of Helsinki statements and was approved by the Ethics Committee of the Sapienza University of Rome, Policlinico Umberto I, Rome, Italy.

Based on the clinical manifestations and medical history, on physician judgment, the following investigations were requested: (a)Antinuclear antibodies (ANA) and anti-dsDNA, detected by means of indirect immunofluorescence (IIF);(b)RF, ACPA and Extractable Nuclear Antigen antibodies (ENA) detected by using commercial ELISA kits (results evaluated according to the manufacturers’ instructions);(c)C3 and C4 serum levels by nephelometry;

Furthermore, according with clinical phenotype, we performed musculoskeletal ultrasound, according to the EULAR guidelines. 

### Statistical Analysis

The statistical analyses were performed using version 5.0 of the GraphPad statistical package. Normally distributed variables were summarised using the mean ± standard deviation (SD), and non-normally distributed variables by the median and interquartile range (IQR). Frequencies were expressed by percentage. Univariate comparisons between nominal variables were calculated using the chi-square test or Fisher’s exact test, where appropriate. Two-tailed *p* values were reported; *p* values less than 0.05 were considered significant. 

## 3. Results

The present descriptive study included 32 patients (M/F 16/16, median age 69, IQR 16.5) affected by malignant diseases treated with pembrolizumab or nivolumab. 

In detail, nineteen patients (59.4%) were affected by non-small-cell lung cancer, seven (21.9%) by head and neck squamous cell carcinoma, three (9.4%) by renal cell carcinoma, two (6.2%) by melanoma and one (3.1%) by urothelial carcinoma. Sixteen patients (50%) were treated with nivolumab, the other half with pembrolizumab. Table 1 reports data about the patients evaluated—in detail, we reported clinical and laboratory assessments which contributed to the diagnosis of different rheumatic conditions. Interestingly, we made a specific rheumatic disease diagnosis for 20 patients (62.5%).

According to the 2010 ACR/EULAR criteria [12], eight patients (25%) presenting polyarthritis could be classified as affected by RA, despite the presence of RF and/or ACPA in only half of the cases. One patient was diagnosed with PsA, due to the development of psoriasis, oligo-arthritis and dactylitis [13]. Following the development of inflammatory shoulder pain associated with the elevation of inflammatory biomarkers, six patients (18.7%) received a diagnosis of PMR. According to EULAR classification criteria and management recommendations, we performed ultrasonographic assessment, revealing the presence of subacromial bursitis with effusion at level of bilateral long head of biceps in all the patients [14]. In one of these patients, we found the positivity for ANA (homogenous pattern) and aSSA (titer 250 UI/mL), without other symptoms suspicious for CTDs. 

Five patients presented other symptoms than musculo-skeletal involvement, allowing the diagnosis of CTDs. Two female patients (6.2%) developed leukopenia, photosensitivity and skin manifestations (malar or subacute rash). The laboratory assessment revealed the positivity for ANA in both cases, and anti-SSA in one patient. Thus, according to 2019 EULAR/ACR criteria [15], a diagnosis of SLE was made for both patients. Furthermore, in two other female patients we observed the presence of sicca syndrome with modification in the Schirmer test, associated with anti-SSA positivity. Accordingly, they received a diagnosis of SjS [16]. In detail, one patient was affected by head and neck squamous cell carcinoma but was not treated by radiation therapy. Finally, in one individual we made a diagnosis of undifferentiated connective tissue disease (UCTD), due to the presence of ANA positivity, leukopenia and purpura on the lower limbs (other possible medical conditions for this manifestation were ruled out). In the remaining patients, eleven subjects have been considered as affected by UA, as they did not meet specific classification criteria; finally, one subject referred only the presence of inflammatory arthralgias and the laboratory exams showed positivity for both ANA (1:80, homogenous) and RF. In this subject, ultrasonographic assessment did not reveal the presence of inflammatory modifications. The median interval time between ICIs starting and the onset of symptoms resulted equal to 14 weeks (IQR 19.75) and was graphically represented in Figure 1. Interestingly, the interval was very low for patients developing CTDs, which was equal to 2 weeks for all the patients except for one SjS subject, showing an interval of 6 weeks. For RA patients, regardless of antibody positivity, we found a median interval equal to 12 weeks (IQR 18), for UA equal to 16 weeks (IQR 32) and for PMR equal to 14 (IQR 10). Furthermore, Table 1 included data about treatment.

At the first rheumatological evaluation, 28 patients were treated by prednisone, at a mean dosage of 11.9 mg/daily (±4.9 SD). The longitudinal observation revealed that RA, PsA and CTD patients required the introduction of treatment with DMARDs, while among UA patients only one needed second line treatment with sulfasalazine. Of course, in the context of the oncology/rheumatology collaboration, the decision to add other than glucocorticoids treatment was made by mutual agreement. 

### Follow-Up

All the patients continued the treatment with ICIs regardless of the rheumatological manifestations. Indeed, concerning Rh-irAEs, 96.5% of patients showed a good response according to physician opinion. From an oncologic point of view, the patients were treated by ICIs for a mean period of 19.7 months (SD 14.02). According to the iRECIST criteria, we observed a complete response in 13.6%, a progression of disease in 50% and death in 22.8%. Finally, 13.6% are still treated by ICIs [17].

## 4. Discussion

The introduction of ICIs substantially changed the history of patients affected by malignant diseases but showed also that the immune system stimulation could induce the development of irAEs and, among these, of Rh-irAEs [2,5]. The growing use of ICIs in a real-life setting confirmed the possible development of different rheumatological conditions, further emphasising the need for shared oncology/rheumatology management of these patients. Indeed, in the present single-centre descriptive study we described a cohort of ICIs treated patients developing a wide spectrum of Rh-irAEs including not only inflammatory arthralgia/arthritis, but also different CTDs. Of note, none of these patients had previously referred signs and symptoms suspicious for rheumatic diseases. 

The heterogeneity of diseases potentially developing during ICIs treatment underlines the relevance of a combined oncologic/rheumatologic evaluation to perform an early rheumatological diagnosis and to introduce the most appropriate treatment. Indeed, in the present cohort, the patients diagnosed with RA and CTD required the introduction of a second line treatment with DMARDs, to control disease and reduce glucocorticoids dosage. Conversely, in patients classified as affected by UA, the treatment with glucocorticoids or NSAIDs was able to induce a prompt and persistent remission of ICIs-induced arthritis. As expected, PMR patients were treated by glucocorticoids in all the cases except one subject, requiring additional treatment for the purpose of steroid sparing. 

Interestingly, it has been possible to make a rheumatologic diagnosis in almost all the patients referred to our attention, underlining that an established collaboration between rheumatologists and oncologists could allow the prompt identification of Rh-irAEs. In a previous analysis, the rate of true inflammatory disease in patients referred to rheumatologist for arthralgia seemed to be lower, probably due to differences in patient referrals [18,19]. Our team has been collaborating with fellow oncologists since 2017 and this ongoing collaboration has certainly improved the ability to select patients, allowing a more targeted patient’s referral [20]. 

With regards to autoantibody status, most of the patients (59.4%) were seronegative for RF or ACPA, in line with previous studies [5,8]; however, a significant proportion of subjects had one or more positivity, including low-titre ANA, associated with the presence of specific clinical manifestations. This evidence would suggest the possible implication of different pathogenic mechanisms. Thus, on one side, the induction of Rh-irAEs might be driven by activated autoreactive T cells; on the other hand, traditional patterns of autoimmunity, probably B cells driven, could be involved, leading to the development of “classical” autoimmune diseases, with a presence of autoantibodies [21].

The development of CTDs represents an interesting topic, which underlines as ICIs treatment was not associated only with the occurrence of arthritis, but also of more complex systemic autoimmune diseases. In particular, several studies demonstrated the possible development of SLE after treatments with some drugs. In this view, ICIs-induced SLE has emerged with the growing use of these drugs. In particular, data from the FDA Adverse Event Reporting System reported the occurrence of SLE in 18 patients among more than 4000 subjects treated [22]. This evidence, together with our results, suggest that ICIs should be added to the list of drug-induced SLE. 

Finally, the interval between the ICIs introduction and the onset of rheumatological manifestations is certainly a crucial factor to consider. According to previous data, in our cohort we found a median interval of 14 weeks [23]. Interestingly, this interval was lower when considering patients developing CTD, in which this interval drops to two weeks. This evidence would suggest a different pathogenic mechanism, but certainly further evidence is needed to confirm this suggestion. 

In conclusion, the heterogeneity of rheumatic conditions which potentially could develop in patients treated by ICIs, certainly supports the need to include the rheumatologist in the management of these subjects. Further studies with large cohorts and longitudinal assessment are needed to identify subjects at risk to develop specific Rh-irAEs.

## Figures and Tables

**Figure 1 life-13-00877-f001:**
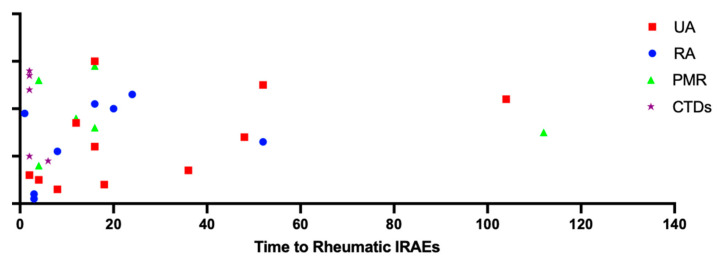
Interval between ICIs introduction and the onset of muscolo-skeletal manifestations.

**Table 1 life-13-00877-t001:** Demographic, clinic and laboratory features and treatment of the patients with ICIs-induced Rh-irAEs.

Pt	Sex	Age	Malignancy (Treatment)	Clinical Manifestations	Interval (Weeks)	Autoantibody Assessment	Diagnosis	Treatment
1	F	55	RCC (nivolumab)	Symmetric polyarthritis	3	RF, ACPA, ANA neg	Seronegative RA	PDN 12.5 mg/daily, HCQ 200 mg bid
2	F	61	Melanoma (nivolumab)	Symmetric polyarthritis	3	RF 22 UI/mL, ACPA >300 UI/mL, ANA + (sp), a-SSA +	RA	PDN 10 mg/daily MTX 10 mg/weekly
3	M	68	NSCLC (nivolumab)	Monoarthritis	8	RF, ACPA, ANA neg	UA	NSAIDs
4	F	72	NSCLC (nivolumab)	Polyarthritis	18	RF, ACPA, ANA neg	UA	PDN 12.5 mg/daily
5	M	77	NSCLC (nivolumab)	Oligoarthritis	4	RF, ACPA, ANA neg	UA	NSAIDs
6	M	70	NSCLC (nivolumab)	Symmetric polyarthritis	2	RF, ACPA, ANA neg	UA	PDN 10 mg/daily
7	M	61	NSCLC (nivolumab)	Symmetric polyarthritis	36	RF, ACPA, ANA neg	UA	PDN 10 mg/daily
8	M	70	HNSCC (nivolumab)	Inflammatory shoulder pain	4	ANA + (h), a-SSA 250 UI/mL	PMR	PDN 10 mg/daily
9	F	80	HNSCC (nivolumab)	Arthralgia, sicca syndrome	6	RF +, ANA + (h), a-SSA 276 UI/mL	SjS	HCQ 200 mg/daily
10	M	74	HNSCC (nivolumab)	Arthralgia	2	RF +, ANA + (h)	Inflammatory arthralgia	NSAIDs
11	F	72	UC (pembrolizumab)	Arthralgia, lymphopenia, porpora	2	ANA + (h),	UCTD	PDN 10 mg/daily, HCQ 200 mg/daily
12	M	59	NSCLC (pembrolizumab)	Polyarthritis	8	ANA, FR, ACPA neg	Seronegative RA	PDN 25 mg/daily
13	M	65	HNSCC (pembrolizumab)	Oligoarthritis	16	ANA ++ (h)	UA	PDN 10 mg/daily
14	F	60	NSCLC (pembrolizumab)	Polyarthritis	52	ANA + (h)	Seronegative RA	PDN 10 mg/daily, MTX 10 mg/weekly
15	M	53	HNSCC (nivolumab)	Oligoarthritis	48	ANA, FR, ACPA neg	UA	PDN 12.5 mg/daily
16	F	78	NSCLC (nivolumab)	Inflammatory shoulder pain	112	ANA, FR, ACPA neg	PMR	PDN 10 mg/daily, SSZ 500 mg tid
17	M	57	NSCLC (pembrolizumab)	Oligoarthritis + psoriasis	78	ANA, FR, ACPA neg	PsA	PDN 25 mg/daily, SSZ 500 mg tid
18	M	85	NSCLC (pembrolizumab)	Inflammatory shoulder pain	16	ANA + (h), FR, ACPA neg	PMR	PDN 10 mg/daily
19	M	80	NSCLC (pembrolizumab)	Polyarthritis	12	ANA, FR, ACPA neg	UA	PDN 12.5 mg/daily
20	M	75	HNSCC (pembrolizumab)	Inflammatory shoulder pain	12	ANA, FR, ACPA neg	PMR	PDN 5 mg/daily
21	M	74	HNSCC (pembrolizumab)	Polyarthritis	1	FR+, ANA + (sp); ACPA 338 UI/ml	RA	PDN 10 mg/daily MTX 10 mg/weekly
22	F	78	Melanoma (pembrolizumab)	Polyarthritis	20	FR neg, ANA neg, ACPA +	RA	PDN 10 mg/daily, SSZ 500 mg bid, HCQ 200 mg bid
23	F	59	NSCLC (pembrolizumab)	Polyarthritis	16	RF +, ANA + (sp), ACPA neg	RA	PDN 10 mg/daily, MTX 10 mg/weekly
24	F	81	NSCLC (nivolumab)	Oligoarthritis	104	ANA + (h), RF, ACPA neg	UA	PDN 10 mg/daily
25	M	64	RCC (nivolumab)	Polyarthritis	24	ANA, FR, ACPA neg	Seronegative RA	PDN 10 mg/daily, MTX 10 mg/weekly
26	F	78	NSCLC (nivolumab)	Polyarthritis + sicca syndrome	2	ANA + (sp), a-SSA 1633 UI/mL	SdS	PDN 12.5 mg/daily, HCQ 200 mg/daily
27	F	62	RCC (nivolumab)	Oligoarthritis	52	ANA, FR, ACPA neg	UA	PDN 10 mg/daily, SSZ 500 mg bid
28	F	85	NSCLC (nivolumab)	Inflammatory shoulder pain	4	ANA, FR, ACPA neg	PMR	PDN 10 mg/daily
29	F	61	NSCLC (pembrolizumab)	Polyarthritis + photosensibility, malar rash, thrombocytopenia	2	ANA +, a-SSA +	SLE	PDN 10 mg/daily, HCQ 200 mg bid
30	F	52	NSCLC (pembrolizumab)	Polyarthritis, subacute rash	2	ANA 1:160 (h)	SLE	PDN 25 mg/daily, HCQ 200 mg bid
31	M	61	NSCLC (pembrolizumab)	Inflammatory shoulder pain	16	ANA, FR, ACPA neg	PMR	PDN 10 mg/daily
32	F	56	NSCLC (pembrolizumab)	Oligoarthritis	16	ANA, FR, ACPA neg	UA	PDN 10 mg/daily

RCC: renal cell carcinoma; NSCLC: non-small-cell lung cancer; HNSCC: head and neck squamous cell carcinoma; UC: urothelial carcinoma; RF: rheumatoid factor; ACPA: anti-citrullinated protein antibodies; ANA: anti-nuclear antibodies (h: homogeneous, sp: speckled); RA: rheumatoid arthritis; UA: undifferentiated arthritis; PMR: polymyalgia rheumatica; PsA: psoriatic arthritis; SjS: Sjögren syndrome; SLE: systemic lupus erythematosus; UCTD: undifferentiated connective tissue disease; LHB: long head of biceps; NA: not available; PDN: prednisone; HCQ; hydroxychloroquine; SSZ; sulfasalazine; MTX: methotrexate.

## Data Availability

Data are available upon reasonable request.

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
