# Peer review of "Rheumatic Diseases Development in Patients Treated by Anti-PD1 Immune Checkpoint Inhibitors: A Single-Centre Descriptive Study"

_life, 2023, doi:10.3390/life13040877_

Round 1
Reviewer 1 Report
This is a descriptive paper that enumerates the patients that were seen in this rheum/onc clinic. My comments:
1. They say in the text that Table 1 includes ultrasound findings, but I did not see any ultrasound findings. Would recommend adding ultrasound findings where relevant
2. I think the paper could be much more useful to the community if the authors include data on a. response of the AE to the treatment; b. length of time of follow up for each patient; c. oncologic outcome
3. In the discussion section the authors reference the rheum/onc collaboration in several places, but I think a brief description of the logistics of such an arrangement may be helpful--it would allow other interested parties to potentially mimic this arrangement.
Author Response
Referee 1
This is a descriptive paper that enumerates the patients that were seen in this rheum/onc clinic. My comments:
- They say in the text that Table 1 includes ultrasound findings, but I did not see any ultrasound findings. Would recommend adding ultrasound findings where relevant.
Response: We thank for this suggestion. However, as mentioned in the Methods Section, we decided to perform ultrasonographic (US) assessment according to the clinical phenotypes of patients and according to the physician opinion. Thus, as reported in the Results section, we performed US evaluation in all the patients suspicious for polymyalgia rheumatica, based on the inclusion of these imaging techniques in EULAR classification criteria. These results were already described. In the other patients, we performed US assessment only in selected cases when clinical and laboratory evaluation were not sufficient to perform a diagnosis. Accordingly, we corrected the title of Table 1.
- I think the paper could be much more useful to the community if the authors include data on a. response of the AE to the treatment; b. length of time of follow up for each patient; c. oncologic outcome
Response: Thank you for this suggestion. All the patients continued the treatment with ICIs regardless of the rheumatological manifestations. Indeed, concerning Rh-irAEs, 96.5% of patients showed a good response according to physician opinion. From an oncologic point of view, the patients were treated by ICIs for a mean period of 19.7 months (SD 14.02). According to the iRECIST criteria,, we observed a complete response in 13.6%, a progression of disease in 50% and death in 22.8%. Finally, 13.6% are still treated by ICIs. These data were added to the Results section and accordingly we modified the References section.
- In the discussion section the authors reference the rheum/onc collaboration in several places, but I think a brief description of the logistics of such an arrangement may be helpful--it would allow other interested parties to potentially mimic this arrangement.
Response: Thank you for this suggestion. Given that oncologic data about Rh-irAEs are very heterogeneous, with arthralgias (irrespective of inflammatory or not) and arthritis often considered overlapping, our established collaboration with oncologists has improved the ability to select patients, allowing a more targeted patient’s referral. Indeed, guiding symptoms that were accurately researched from oncologists in order to refer the patients to our attention were inflammatory arthralgias, arthritis and/or other manifestations suspicious for rheumatic diseases (photosensitivity, malar rash, sicca syndrome, Raynaud phenomenon, psoriasis, aphthosis, serositis, haematological modifications, uveitis, purpura, thrombotic events). Moving to the treatment, the decision to add an immunosuppressant drug as DMARD, if necessary, was made in agreements with oncologists, pointing out that treatment of neoplasm is certainly always prioritized. These new information were added in the text.

Reviewer 2 Report
The manuscript is well-organized and clearly stated.This study contains some interesting findings and are valuable for the understanding of rheumatic conditions developing during treatment with anti-PD1 immune-checkpoint inhibitors. However,lack of Sufficient clinical samples is the major flaw of the study.Therefore,MINOR revision has to be done before this manuscript could be accepted for publication in the LIFE.I would suggest accepting it after the following minor concerns are addressed.
1.In the introduction section, the authors need to provide detailed information on current progress in anti-PD1 immune-checkpoint inhibitors.
2.The data in Fig1 are not sufficient to demonstrate that the interval between ICIs initiation and the onset of musculoskeletal symptoms is very low for patients developing CTDs. If you have more samples, the results will be more meaningful.
Author Response
Referee 2
The manuscript is well-organized and clearly stated. This study contains some interesting findings and are valuable for the understanding of rheumatic conditions developing during treatment with anti-PD1 immune-checkpoint inhibitors. However, lack of Sufficient clinical samples is the major flaw of the study. Therefore, MINOR revision has to be done before this manuscript could be accepted for publication in the LIFE.I would suggest accepting it after the following minor concerns are addressed.
1. In the introduction section, the authors need to provide detailed information on current progress in anti-PD1 immune-checkpoint inhibitors.
Response: We thank you for this suggestion and we modified the Introduction section.
2.The data in Fig1 are not sufficient to demonstrate that the interval between ICIs initiation and the onset of musculoskeletal symptoms is very low for patients developing CTDs. If you have more samples, the results will be more meaningful.
Response: Although the sample size of ICIs-associated CTDs is low and the difference between these Rh-irAEs onset and the others was not statistically significant, we consider interesting their onset after only 2 weeks from the treatment start. We hope that the ongoing collaboration with oncologists and the growing use of different ICIs could increase the sample size.

Round 2
Reviewer 1 Report
No further comments